# Transconjunctival and Subciliary Approach in the Treatment of Orbital Fractures: A Study on Oculoplastic Complication

**DOI:** 10.3390/jcm10132775

**Published:** 2021-06-24

**Authors:** Lorenzo Trevisiol, Antonio D’Agostino, Silvia Gasparini, Pierfrancesco Bettini, Massimo Bersani, Riccardo Nocini, Vittorio Favero

**Affiliations:** 1Unit of Maxillo-Facial Surgery and Dentistry, University of Verona, P.le L.A. Scuro 10, 37134 Verona, Italy; Lorenzo.trevisiol@univr.it (L.T.); silvia.gasparini@univr.it (S.G.); maxibers@hotmail.it (M.B.); vittorio_favero@yahoo.it (V.F.); 2Unit of Otolaryngology, University of Ferrara, Via Savonarola 9, 44121 Ferrara, Italy; pierfrancescobettini@gmail.com; 3Unit of Otolaryngology, University of Verona, P.le L.A. Scuro 10, 37134 Verona, Italy; riccardo.nocini@gmail.com

**Keywords:** oculoplasty, transconjunctival, subciliary, complication

## Abstract

(1) Background: The aim of the present study was to compare lower eyelid post-operative complications, such as ectropion, entropion, and scleral show of orbital floor fractures, associated to the subciliary vs transconjunctival approaches. (2) Materials and Methods: A retrospective comparative study of patients who underwent surgery for orbital fractures by means of a transconjunctival or a subciliary approach at the Clinic of Dentistry and Maxillofacial Surgery of the University of Verona from January 2013 through September 2018 was designed. Data related to the trauma and to surgical procedures were retrieved, as well as a series of anthropometric parameters extrapolated from standardized photographs. Statistical analysis was performed on the outcomes. (3) Results: 33 patients underwent surgery by means of a transconjunctival approach and 36 patients by means of a subciliary approach. Ectropion was observed to a greater extent in the subciliary group, however the difference resulted to be not statistically significant. Patients in which osteosynthesis devices were used presented with a greater incidence of scleral show with respect to the remaining patients. No statistically significant difference was observed for any of the parameters taken into account. (4) Conclusions: Since the two approaches does not seem to be associated with remarkable differences in terms of outcomes, the choice of technique should be tailored to the patient’s features and the surgeon’s experience.

## 1. Introduction

The face is an extremely complex area given its functional and aesthetic importance. Traumas of the face can determine soft and hard tissue injuries necessitating surgical intervention. While there is more than one surgical approach that can be used to repair orbital floor fractures and restore normal anatomy and function [1], the transconjunctival and transcutaneous are the most frequently utilized ones to access the lower eyelids to treat orbital fractures. Both have, in fact, been utilized by surgeons for several decades to treat periocular and ocular injuries [2,3,4,5]. There is currently, however, no consensus in the literature about which approach is linked to the best aesthetic-functional results and minimizes or prevents the complications linked to lower eyelid surgery. Achieving a good exposure of the orbital floor while minimising scarring continue to be primary considerations, and clearly much depends on the skill and experience of the surgeon. All incisions performed near the lower eyelid are subject to the low but the nevertheless possible risk of complications such as ectropion, entropion, lower eyelid retraction or malpositioning, all conditions that can be serious and debilitating both functionally and aesthetically [6]. While the transconjunctival approach can best conceal signs of fornix conjunctival incision leaving no visible scarring and provide a good exposure of the orbital floor, it is more frequently linked to entropion and malpositioning especially in those cases in which a further incision such as a canthotomy is needed to widen the operating field. However, although the transcutaneous approach offers a better exposure of the surgical site without interrupting the medial canthal tendon, it is more frequently associated to lagophthalmos as well as to a higher incidence of ectropion, eyelid retraction, and visible scarring [7]. This study’s primary aim was to compare lower eyelid post-operative complications such as ectropion, entropion and scleral show of orbital floor fractures associated to the subciliary vs transconjunctival approaches in two groups of patients. Its secondary aim was to investigate factors, such as the type of orbital fracture and their causes, the use of osteosynthesis devices and recourse to canthotomy during surgery, and their relationships with the two types of surgical approaches.

## 2. Materials and Methods

A retrospective, comparative study was carried out by reviewing the hospital records of the patients treated at the Oral-Maxillofacial Surgery Unit of the University of Verona Medical Center because of orbital fractures, such as blow-out, orbito-zygomatic (OZ), or naso-orbital-ethmoid (NOE) ones, between January 2013 and September 2018. Patients who underwent surgery to repair orbital facial fractures in which the pre-septal transconjunctival (Figure 1a–c) or subciliary (Figure 2a–c) approach was used, who were monitored post-operatively for at least a year’s time, and whose clinical and photographic records were complete were enrolled in the study. The surgical interventions were performed by three different consultants.

Subjects who had undergone a previous or who underwent further procedures of the lower eyelid, who had a previous eyelid trauma, who presented traumatic lesions of the eye bulb or of the soft tissues of the eyelid or any thyroid pathology were excluded. All of the patients who met the inclusion criteria were contacted and were provided information about the study’s aims and methodologies. They were asked to sign consent statements showing that they agreed to participate in the study and giving the investigators permission to review their charts. Those who agreed underwent a comprehensive clinical examination and their photographic records were reviewed to investigate the presence of any signs of surgery-related complications. Standard photographs were taken by a professional photographer standing a meter away from the patient using a Canon EF 100 mm EOS 5D camera (Canon EF Macro lens 1:2.8 D USM with advanced ultrasound focusing and Sunpak GX8R ring flash) with a 250 W flash (the pilot light is equipped with two lateral diffuser screens placed at a 40° angle with respect to it). The photographs were taken with a millimetric scale positioned at the side of the patient’s face to permit the investigators to analyze and compare the measurements. The surgery-related variables analyzed were: ectropion, entropion, and the scleral show. The other variables that were analyzed were: the patient’s anagraphic features (age, sex, the length of time he/she was monitored post-operatively, the cause of the injury), if a canthotomy was required and if osteosynthesis devices were used. In those cases in which the patient underwent bilateral treatment, each eye was evaluated independently. Scleral show was evaluated by measuring the anthropometric features extrapolated from the photographs using a standardized millimeter scale present in the photos and comparing the operated eye with the contralateral one. Scleral show, which was considered significant if it was greater than 1 mm, was evaluated only in the patients who had monolateral fractures since a comparison between the two eyes of a patient with bilateral fractures would be irrelevant in the absence of standardized photographs taken precedent to the trauma. Only the quantifiable measurements of the patients with eyelid ptosis or dermatochalasis of the upper eyelid impairing vision of the most lateral corner of the iris were investigated. The following measurements were determined using the GIMP-2.10.12 program, and the anthropometric data were analyzed (Figure 3).

The iris vertical visible (VI) height (in mm), that is the vertical straight line (diameter) passing through the center of the pupil uniting the points of the upper and lower eyelid rims from which a horizontal reference line could be drawn.The lower iris radius visible (RI) height (in mm), that is half of the VI uniting the center of the pupil to the point on the rim of the lower eyelid from which a horizontal reference line could be drawn.The iris margin (MI) height (in mm), that is the distance on a vertical reference line drawn at the most lateral corner of the iris between the lower and upper eyelids.The medial-lateral (ML) height (in mm), passing perpendicularly to the middle point of the horizontal line between the most lateral corner of the iris and the intersection point of the line with the vertical line passing through the lateral canthus. It unites the points of the interior and superior eyelid rims on a reference line.The eye fissure height (EFH) (in mm) [8], that is the vertical distance from the margin of the inferior lid to the superior one.The eye fissure width (EFW) [8], that is the horizontal distance between two vertical lines passing through the medial and lateral canthus, defined by the intercanthal distance (in cm).The eye fissure index (EFI), defined as the EFH divided by the EFW. This measurement is the preferred one with respect to the EFH and the EFW because it is reproducible in photographs taken in standardized positions with respect to linear measurements.

The mean difference between the two eyes of the anthropometric measurements was calculated. It was considered positive if the measurements of the operated eye were greater those of the contralateral eye and negative if those of the non-operated eye were greater than its counterpart. The mean differences between the approaches were compared and analyzed.

### Statistical Analysis

The χ2 and Fisher’s exact tests were carried out to compare: the rates of ectropion linked to the two approaches, the association between ectropion and the use of osteosynthesis devices, and the association between ectropion and the type of eye fracture. The Shapiro-Wilk test was used to verify the fit of the data to the normal distributions of each continuous variable of the anthropometric measurements to evaluate scleral show. The Wilcoxon-Mann-Whitney test was used to compare the mean difference between the two eyes of each anthropometric measurement with reference to: the two approaches, the use of osteosynthesis devices and recourse to canthotomy. The Kruskal-Wallis test was carried out to compare the mean difference between the two eyes for each anthropometric measurement in relation to the three types of fractures studied. The calculations were made using the SPSS V 19.0 software for Windows. A *p*-value of <0.05 was considered significant.

## 3. Results

Sixty-nine (16 were female and 53 were male; mean age was 42 years with a range between 6–78) patients met the study’s inclusion criteria. The mean follow-up time was 35 months (range = 12–74 months). Twenty-one patients (30%) presented blow-out fractures, 40 (58%) presented OZ fractures, and eight (12%) presented NOE fractures. The breakdown of the causes of the orbital osseous traumas was: motor vehicle accidents (31 patients = 45%), assaults (seven patients = 10%), accidents (22 patients = 32%), sports-related accidents (nine patients = 13%) (Table 1).

There were no cases of entropion in either of the two patient groups. There were three out of 37 (8.11%) cases of ectropion associated to the subciliary approach but none (0 in 37 = 0%) to the transconjunctival one. The difference with regard to the latter was not statistically significant (*p* = 0.077) according to the χ2 test, a result that was confirmed by Fisher’s exact test (Table 2).

The mean differences between the two eyes of the anthropometric measurements to evaluate the scleral show were compared, but none were statistically significant (Table 3).

No statistically significant associations were found between entropion, ectropion or scleral show, and the type of fracture or recourse to canthotomy, nor were there any correlations with the type of access utilized (Table 4 and Table 5).

There was, instead, a statistically significant association between the use of osteosynthesis devices and the MI, ML, and EFI measurements (Table 6). The Wilcoxon-Mann-Whitney test was utilized to compare the two surgical approaches, but no statistically significant results were uncovered.

## 4. Discussion

The current study set out to compare some of the lower eyelid complications, such as ectropion, entropion, and scleral show of orbital floor fracture surgery associated to the subciliary vs transconjunctival approaches to access the orbital floor in the effort to investigate which produces better aesthetic and functional results at the end of a one-year follow-up period. The first result emerging from our analysis was that none of the patients in either group presented entropion. Instead, although the correlation was not statistically significant, the cases of ectropion were found only in the transcutaneous-treated patients. As far as scleral show was concerned, according to our evaluation, which was based on anthropometric measurements taken in the effort to be as objective and reproducible as possible [9], there did not seem to be any substantial differences between the two groups. A more detailed narration of our analysis should include that the only anthropometric measurement linked to the subciliary approach that was associated to a greater, although not statistically significant, enlargement of the operated eye with respect to its counterpart was the iris radius visible (RI). Consistently with the values of all the other anthropometric measurements for the two approaches, the RI of the operated eye following the transcutaneous approach tended more frequently with respect to the contralateral eye to be associated with scleral show. On the other hand, the transconjunctival approach on average showed a reduced vertical aperture of the operated eye with respect to that of the contralateral eye. As far as the ML parameter was concerned, following the transconjunctival approach the operated eye presented higher, but never statistically significant values with respect to the contralateral eye. This could be explained by the scarring linked to canthotomy as 80% of the fractures accessed using the transconjunctival approach required a canthotomy. It was found that the mean ML measurement of the operated eye was higher with respect to that of the contralateral eye and that the difference between the two eyes was greater with respect to that found in the non-canthotomy patients. There was virtually no difference between the two eyes in the non-canthotomy patients. The difference in the ML value between the two groups was not statistically significant, perhaps because of the small number of patients studied. When the anthropometric measurements to evaluate scleral show in the groups stratified according to the type of fracture they presented (OZ, NOE, or blow-out) were analyzed, it was found that there were differences between the OZ + NOE values with respect to the blow-out ones. In fact, the mean differences between the two eyes for all five anthropometric values (VI, RI, EFI, MI, ML) were minor in the operated eye in the blow-out group with respect to the contralateral one (negative differences), and there were overlapping differences for both the central and lateral measurements of the lid. The operated eye in the OZ and NOE groups was, instead, larger with respect to the contralateral one. More than one factor may have played a role here: first of all, the type of fracture. Blow-out fractures cause more disruption of the orbital soft tissues and are linked to a higher incidence of post-traumatic enophthalmos even when bone reconstruction is appropriate [10]. Secondly, fibrotic scarring connected to the use of osteosynthesis devices, positioned exclusively in the OZ and NOE fractures, could explain a retraction effect, perhaps at the subclinical level, of the palpebral tissues. In accordance with this hypothesis, the differences in the various anthropometric measurements between the osteosynthesis and non-osteosynthesis groups tended to be similar. There was a statistically significant difference in the EFI, MI, and ML in both groups, with greater differences between the two eyes in the patients in whom osteosynthesis devices were used. The results regarding RI showed a similar pattern although they did not reach statistical significance. It is important to note that the same percentage of osteosynthesis devices were used during the two surgical approaches. It is likewise noteworthy that osteosynthesis devices were not associated with a statistically significant difference as far as ectropion was concerned, which indicates that the results regarding the scleral show may have referred to a pre-clinical condition. The articles in the literature that have compared the transcutaneous and transconjunctival approaches to the infraorbital rim/orbital floor and their relationships with ectropion, entropion, and scleral show have not reached a consensus about the best way to manage orbital fractures. According to a systematic review and meta-analysis by Al-Moraissi et al. [11], the transcutaneous approach, in particular the subciliary one, is associated to a higher incidence of ectropion, both with regard to its permanent form (which requires surgical treatment) as well as to its transitory form (which is managed in a conservative manner). The finding concords with the data of some studies, such as those by Pospisil and Fernando [12], Friesenecker et al. [13], Patel et al. [14], Lorenz et al. [15], de Melo Crosara et al. [16], although other reports have demonstrated a lower incidence of ectropion with respect to that described by some meta-analyses [6,17,18,19]. The ectropion associated to the subciliary approach could be due to post-operative edema that could push the lower palpebra away from the eye bulb or to tightening with or without scarring of the anterior and middle lamella. It could also be due to the fact that the anterior lamella, including the skin and the orbicularis muscle along the preseptal plane, was dissected. Al-Moraissi’s meta-analysis [11] concluded that the transconjunctival approach should be considered superior not only because scarring is concealed in the interpalpebral conjunctiva but also because it is associated to a statistically significant lower incidence of ectropion with respect to the subciliary approach. The authors acknowledged however that it was accompanied by a higher incidence of entropion [6,17,20,21,22,23,24,25]. The higher incidence of entropion that was noted could be explained by the fact that the posterior lamella, which is made up of the conjunctiva, the tarsal plate, and the eyelid retractors of the inferior palpebra, was dissected. This made it possible to save the orbicularis oculi muscle fibers, and to avoid dissection across the anterior lamella (only the lateral canthus was involved), which could explain the lower incidence of ectropion [26]. It should also be remembered that the transconjunctival approach can ensure access to the lateral orbital wall only if a lateral orbital canthotomy is performed, something that can cause scarring and eyelid mispositioning. These observations are consistent with those of other studies [20,27,28,29]. The results outlined in the literature reporting a higher incidence of ectropion in the patients who underwent the transcutaneous approach with respect to their transconjunctival counterparts are partially in agreement with ours although the difference in our patients was not statistically significant. The differences between our results and the literature could be related to the relatively limited size of the study population together with the strict inclusion criteria adopted. Further large-scale studies are thus warranted. Another study limitation was its retrospective nature as bias may have been introduced in the selection of the subjects as well as in the planning, data collection and analysis, and interpretation phases. Moreover, it was objectively impossible to compare our palpebral measurements with pre-trauma ones.

## 5. Conclusions

To conclude, an analysis of the study’s results has demonstrated that the post-operative outcomes of orbital fracture procedures accessed using the transconjunctival or subciliary approaches in two groups of patients were analogous. In the light of that result, the decision to opt for one or the other approach should be based entirely on the features of the patient requiring surgery, bearing in mind not only possible lower eyelid complications, but also clinical and cosmetic considerations, probable surgical times, the experience and skill of the surgeon, the difficulty of the operation, the most efficacious access to exposing the orbital floor to repair the fracture, and the importance of avoiding visible scarring that could ruin the harmony and aesthetics of the patient’s face. The transconjunctival approach could be opportune if the fracture involves the orbital floor and/or wall in order to ensure that scarring is avoided. A transcutaneous approach may be more advisable if the probable surgical time is limited, and in the case palpebral lesions, old scars or skin flaps are involved. In the case of the authors, performing the present study definitely yielded more awareness on the advantages and disadvantages of either approach. This will hopefully influence in a positive manner the outcomes of future surgeries. Future large prospective studies will be able to provide more information about the pros and cons of the two approaches and how the type of fracture being repaired, the use of osteosynthesis devices, and recourse to canthotomy can affect the outcome of surgery.

## Figures and Tables

**Figure 1 jcm-10-02775-f001:**
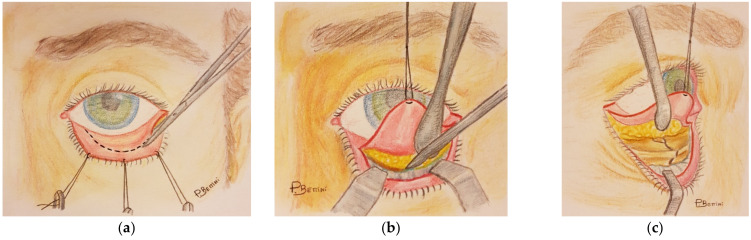
Transconjunctival approach: (**a**) incision (**b**) dissection of arcus marginalis (**c**) fracture exposition.

**Figure 2 jcm-10-02775-f002:**
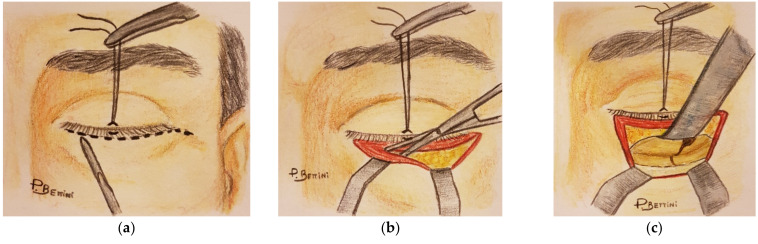
Subciliary approach: (**a**) incision (**b**) preseptal dissection (**c**) fracture exposition.

**Figure 3 jcm-10-02775-f003:**
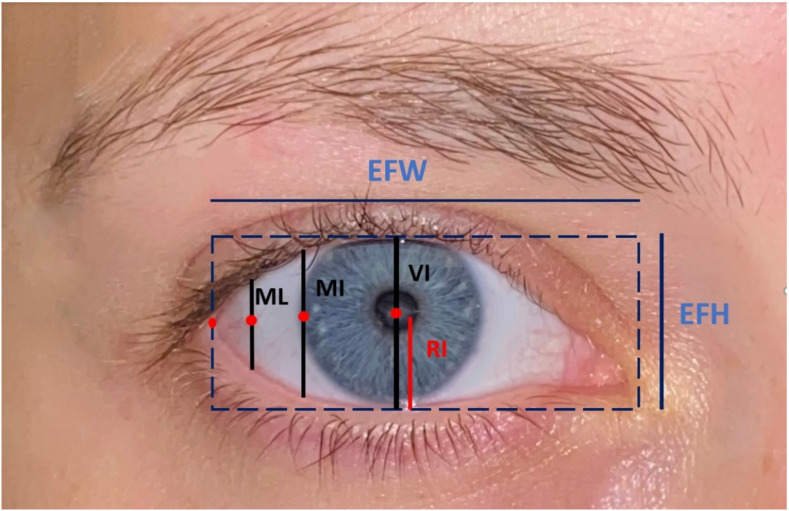
Anthropometric measurements.

**Table 1 jcm-10-02775-t001:** The number, sex, and ages of the patients studied and the mean follow-up times, the type of osseous traumas they presented and their causes and the number of canthotomies performed and fixation devices used in the patients divided into two groups depending on the approach used: transconjunctival or subciliary.

	Transconjuctival	Subciliary
Number of patients	33 *	36 **
Number of operated eyelids	37	37
Sex (n. of patients)		
male	25 (75.76%)	28 (77.78%)
female	8 (24.24%)	8 (22.22%)
Mean age (years)	44 (range: 9–71)	39 (range 6–78)
Mean follow-up time (months)	22 (range: 12–40)	46 (range: 12–74)
Cause of the trauma (n. and % of patients)		
Motor vehicle accidents	15 (45.45%)	16 (44.44%)
Assaults	3 (9.09%)	4 (11.11%)
Accidents	10 (30.3%)	12 (33.33%)
Sports-related accidents	5 (15.15%)	4 (11.11%)
Type of fracture (n. and % of patients)		
Blow-out	11 (33.33%)	10 (27.78%)
NOE	5 (15.15%)	3 (8.33%)
Orbito-zygomatic	17 (51.51%)	23 (63.89%)
Osteosynthesis devices used (n. and % of patients)	26 (70.27%)	26 (70.27%)
Recourse to canthotomy (n. and % of patients)	30 (81.08%)	

* Transconjuctival approach: The case of bilateral orbito-zygomatic was considered a single one as the fractures were evaluated independently. Four of the patients underwent bilateral transconjunctival procedure; in these cases a bilateral comparison of the scleral show was impossible. In four other patients it was difficult to evaluate the photographic records of the scleral show given the anthropometric characteristics of the lower eyelid, therefore in three of them a standardized measurement of only the iris vertical visible height could be made, whereas in the fourth patient both the the lower iris radius visible and the iris vertical visible were measured. ** Subciliary Approach: One of these patients underwent a bilateral transcutaneous procedure in which it was impossible to compare the scleral show bilaterally. Moreover, it was impossible to evaluate the scleral show in one patient who was without photographic records. Finally, it was difficult to evaluate the photographic records of the scleral show of two patients because it was possible, due to the anthropometric characteristics of their lower eyebrows, to take a measurements of only the lower iris radius visible.

**Table 2 jcm-10-02775-t002:** The number of patients presenting entropion and ectropion in the two study groups. A *p* value of <0.05 was considered significant.

		Transconjuctival (Number of Lesions)	Subciliary (Number of Lesions)	*p*-Value
Entropion	no	37 (100%)	37 (100%)	*p* = 1
	yes	0	0	
Ectropion	no	37 (100%)	34 (91.89%)	*p* = 0.077
	yes	0	3 (8.11%)	

**Table 3 jcm-10-02775-t003:** The continuous variables (the anthropometric measurements) to evaluate scleral show in the two patient groups.

Anthropometric Measurements to Evaluate Scleral Show	Transconjuctival (Mean Values)			Subciliary (Mean Values)			*p*-Value
	Operated Eye (cm)	Uninjured Eye (cm)	Δ Between the Two Eyes (mm)	Operated Eye (cm)	Uninjured Eye (cm)	Δ Between the Two Eyes(mm)	
VI (iris vertical visible height)	1.01 ± 0.14	1.0 ± 0.17	0.05 ± 0.87	0.99 ± 0.15	0.96 ± 0.11	0.28 ± 1.2	0.3721
RI (iris radius visible)	0.67 ± 0.14	0.68 ± 0.13	−0.03 ± 0.85	0.7 ± 0.22	0.66 ± 0.22	0.33 ± 1.02	0.0574
MI (iris margin height)	0.85 ± 0.15	0.84 ± 0.15	0.15 ± 0.92	0.79 ± 0.17	0.77 ± 0.12	0.17 ± 1.15	0.8848
ML (mean lateral height)	0.62 ± 0.14	0.59 ± 0.13	0.35 ± 0.86	0.56 ± 0.26	0.56 ± 0.26	0.09 ± 0.85	0.1945
EFI (eye fissure index)	0.44 ± 0.1	0.43 ± 0.07	0.01 ± 0.04	0.42 ± 0.06	0.4 ± 0.06	0.02 ± 0.06	0.2512

**Table 4 jcm-10-02775-t004:** The associations between the anthropometric measurements to evaluate the scleral show and the three types of fractures studied.

Anthropometric Measurements to Evaluate the Scleral Show	OZ Fractures (Mean Δ between the Two Eyes) (mm)	NOE Fractures (Mean Δ between the Two Eyes) (mm)	Blow-out Frattures (Mean Δ between the Two Eyes) (mm)	*p*-Value
VI (Iris vertical visible height)	0.3314286 ± 1.204348	0.2 ± 1.051982	−0.1263158 ± 0.7030427	0.4392
RI (Iris radius visible height)	0.3621622 ± 1.051283	0.3 ± 1.465151	−0.1952381 ± 0.5132993	0.0928
MI (The Iris Margin height)	0.3441177 ± 1.193211	0.3 ± 0.5773503	−0.1894737 ± 0.736278	0.1620
ML (the medial-lateral height)	0.3529412 ± 0.9320019	0.5 ± 0.5773503	−0.1315789 ± 0.664206	0.0986
EFI (Eye fissure index)	0.0294286 ± 0.0538485	0.0125 ± 0.0670199	−0.0021053 ± 0.0367543	0.0726

**Table 5 jcm-10-02775-t005:** The association between the anthropometric measurements and recourse to canthotomy.

Anthropometric Measurements for the Scleral Show	The Cases in Which Canthotomy Was Performed (Mean Δ between the Two Eyes) (mm)	The Cases in Which No Cantotomy Was Performed (Mean Δ between the Two Eyes) (mm)	*p*-Value
VI (Iris vertical visible height)	0.1473684 ± 0.9100479	−0.2285714 ± 0.7454625	0.5817
RI (Iris radius visible height)	−0.1473684 ± 0.9100479	0.0428571 ± 0.4961759	0.2403
MI (The Iris Margin height)	0.2833333 ± 0.879338	−0.1857143 ± 1.009007	0.4857
ML (the medial-lateral height)	0.4611111 ± 0.8282788	0.0571429 ± 0.9306935	0.3936
EFI (Eye fissure index)	0.0178947; 0.0462576	−0.0057143; 0.021492	0.2680

**Table 6 jcm-10-02775-t006:** The associations between the anthropometric measurements for the scleral show and the use of osteosynthesis devices.

Anthropometric Measurements to Evaluate the Scleral Show	The Cases in Which Osteosynthesis Devices Were Used (Mean Δ between the Two Eyes) (mm)	The Cases in Which Osteosynthesis Devices Were Not Used (Mean Δ between the Two Eyes) (mm)	*p*-Value
VI (Iris vertical visible height)	0.3447368 ± 1.181318	−0.155 ± 0.6962116	0.1236
RI (Iris radius visible)	0.36 ± 1.088589	−0.1772727 ± 0.5079668	0.0552
MI (Marginal iris height)	0.3594595 ± 1.147814	−0.2 ± 0.7181848	0.0371
ML (Medial-lateral height)	0.3837838 ± 0.9038539	−0.135 ± 0.6466716	0.0264
EFI (Eye fissure index)	0.0294737 ± 0.0541243	−0.004 ± 0.0367638	0.0200

## Data Availability

The data presented in this study are available on request from the corresponding author.

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
