# Peer review of "Transconjunctival and Subciliary Approach in the Treatment of Orbital Fractures: A Study on Oculoplastic Complication"

_jcm, 2021, doi:10.3390/jcm10132775_

Round 1

Reviewer 1 Report

Periorbital region fractures are the most common in maxillo-facial surgery practice, as shown in many review, and the best way to ensure access and approach the fracture is still controversial due to the particular anatomy that easily leads to aesthetic complications. In this article, the Authors investigate the differences in terms of complications following two surgical access: subciliar and transconjunctival. The study was well conducted with a precise method, fully explained in the article. The parameters considered to study the severity of the different complication are well described, clear, reproducible and easily appliable in every clinical practice. Iconography and table are good and features appropriate captions. Discussion and conclusion chapter are well exposed and reflect fully the obtained result, giving a comprehensive synthesis of them. Written in correct English.

Author Response

Thank you for your appreciation of the manuscript

Reviewer 2 Report

The paper is well written, amount of information adequate, statistical analysis well performed. 

Author Response

(The authors gave the same response as above.)

Reviewer 3 Report

it is a nice study , which regards a topic that will always be object odf discussion. the anthropometric study , the images and data provided are well represented and organized.

the discussion is complete and satisfactory, the conclusions are supported by the data presented.

i would only ask the authors to describe  which were the criteria used to chose a transconjuntcival approach versus a transcutaneous one and whether these criteria will be changed after the study.

Author Response

Thank you for your appreciation of the paper. Given its retrospective nature, we could not properly determine the reason behind the choice of the approach in each case taken into examination; however, we tried to better express in the conclusion section the way this study might influence our future surgeries.